# Technology-Enhanced Learning in Health Sciences: Improving the Motivation and Performance of Medical Students with Immersive Reality

Julio Cabero-Almenara [1] , Fernando De-La-Portilla-De-Juan [2] , Julio Barroso-Osuna [1]
and Antonio Palacios-Rodríguez [1,*]

[1] Department of Didactics and Educational Organization, University of Seville, 41004 Sevilla, Spain;
cabero@us.es (J.C.-A.); jbarroso@us.es (J.B.-O.)
[2] Department of Surgery, University of Seville, 41004 Sevilla, Spain; fportilla@us.es
* Correspondence: aprodriguez@us.es

**Abstract:** Numerous studies suggest that immersive reality (IR) is an educational technology with great potential in the field of health sciences. Its integration allows for an increase in the motivation and academic performance of students. In this sense, this research aims to study the self-perception of motivation and performance levels obtained by students who are completing their degree in medicine at the University of Seville after experiencing a session with IR. To achieve this, 136 student participants answered two questionnaires, the IMMS and the academic performance test. The results show high levels of motivation during the IR session, where the interaction with "hot spots" predominates. In the same way, the measured performance results are quite great. For this reason, it is concluded that the potential of using IR as an educational technology is evident, and new lines of related research are opened.

**Keywords:** immersive reality; educative technology; university education; digital competence; health sciences; emerging technologies



## 1. Introduction

Virtual reality (VR) is one of the technologies that has recently gained greater interest in its incorporation into teaching, especially in higher education or university. Regarding its conceptualization [1], its relation to other similar technologies needs to be pointed out. There is a clear difference between augmented reality (AR) and virtual reality (VR) since, in the latter, virtual data replace physical data, creating a new reality. On the contrary, in augmented reality, the two realities are superimposed on different layers of information in different formats (computer-generated images, video sequences, animations, etc.) to set up a new reality that is the one with which the person interacts. In any case, it cannot be ignored that both "realities" share the following common features, as has already been pointed out: immersion, navigation, and interaction. Immersive reality (IR) is a combination of both technological formats.

According to [2], IR is also defined as an environment that may or may not appear real, which gives the user the feeling of being immersed in it. As a general rule, this environment is generated by a computer system and viewed by the user through a specific device such as a helmet or glasses and, depending on the system and how elaborate and immersive it intends to be, it may be accompanied by other elements such as position and movement sensors, gloves, sound, and elements such as control remotes to move or manipulate objects in the environment.

At the same time, we must consider that within VR, we find immersive and non-immersive VR. As the authors of [3] mention, VR has a series of specific characteristics; thus, immersive VR presents the following: it has a high cost, its use is complex, it tends

to cause disorientation in the person, it offers a great sensation of reality, and it provides a sensation of total immersion. Non-immersive VR presents a more accessible cost; it is easier to use, it offers rapid acceptance, and it expresses a lesser sense of reality and the partial immersion that is achieved with it. Other authors make a distinction between low-immersion virtual reality, which is based on traditional devices such as the mouse and keyboard, and high-immersion reality, which generally involves a head-mounted display [4].

In any case, we have to bear in mind that they are terms that cause a certain amount of confusion [5], as presented below in Figure 1.

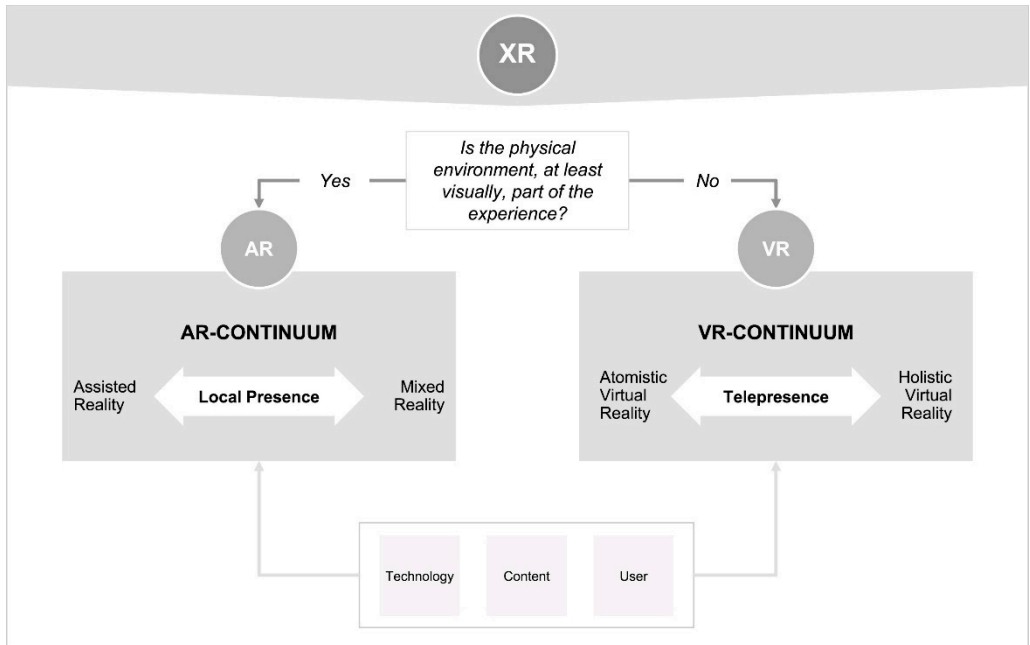

**Figure 1.** Scheme of mixed or extended reality and augmented and virtual reality.

Now, we have to be aware that even if there is an increasing interest in VR, the reality is that there is little research on it. The three main reasons for the lack of studies and research [6] are the following: (1) these technologies are accessed through smartphones, the use of which has not been consolidated in the teaching–learning process; (2) the lack of content on the use of these technologies in educational centers; and (3) the digital divide in countries and schools.

However, lately, especially since 2020, the scientific production regarding this topic has increased [7]. This has resulted in an increase in meta-analyses on studies and research that usually offer positive results and improve academic performance when used in educational contexts [8–11].

Several studies have documented the positive impact of virtual reality (VR) on learning in various contexts [9,12–14]. VR has been found to enhance students' sense of physical presence during training, improve their learning experiences, boost their confidence, and increase their interest in training activities. Consequently, students tend to express higher levels of satisfaction when engaging in training using VR technology. Additionally, VR has shown promising results in the treatment of dyslexia, as evidenced by several studies [15–19].

We also found a large volume of research that points out the relevance of its use in enhancing experiential learning [20–22]. Therefore, it favors overcoming the difference between theory and practice [23] and promotes the creation of more active training scenarios for students [11].

Regarding motivation, numerous studies have demonstrated that the use of virtual reality (VR) enhances students' motivation towards the subject matter and activities they need to engage with [18,22,24,25].

In a related vein, some investigations have focused on optimizing the design of VR technology to improve its performance efficiency [4,26]. Specifically, researchers have explored the effectiveness of incorporating "hot spots", signals, or informative cues to enhance user experience and reduce cognitive load associated with the use of VR [27,28].

The growing adoption of VR in educational institutions can be attributed to several factors, including the increasing capabilities of computers to deliver high-quality audiovisual and multimedia information, as well as the decreasing cost of VR equipment such as headsets. However, it is important to acknowledge that cost, novelty, and complexity still pose limitations to its widespread use.

It is worth noting that VR technology has primarily been utilized in the field of health sciences. Extensive research has demonstrated its effectiveness in simulating various processes and contexts, providing students with realistic experiences and access to challenging environments [12,22,29–33].

In conclusion, the application of VR in education has proven beneficial, particularly in terms of motivating students and creating immersive learning experiences. Ongoing advancements in technology and continued research will likely address the limitations and further expand the potential of VR in education.

## 2. Motivation as a Key Element for Learning and Training

Motivation is widely recognized as a crucial factor for academic success and an essential focus for teachers seeking to engage their students [34,35]. It encompasses the processes that activate, direct, and sustain behavior [36]. Key indicators of motivation include the level of activation, the selection of actions from a range of possibilities, and the concentration of attention and perseverance when facing a task or activity [37]. Additionally, motivation refers to the magnitude and direction of behavior, encompassing a person's choices, goals, and the level of effort they invest [37].

Traditionally, motivation has been categorized into intrinsic (internal) and extrinsic (external) factors according to Deci and Ryan's self-determination theory [37,38]. Information and communication technologies fall under the category of extrinsic variables, and numerous studies have explored their impact on motivation, often reporting increased motivation among students and, in some cases, improved academic performance [39–41].

In the context of teaching, motivation is considered an internal state or condition that influences students' willingness to participate in classroom activities and their reasons for doing so [42]. One widely used model for promoting motivation in instructional design is the ARCS model, developed by [37,43,44]. This model suggests that motivation is determined by four dimensions: attention (A), relevance (R), confidence (C), and satisfaction (S). Attention involves characteristics such as curiosity and the search for sensations. Relevance refers to perceptions of tools that satisfy personal needs and facilitate the achievement of goals. Confidence incorporates various motivational constructs, including perceptions of personal control and the illusion of success. Finally, satisfaction plays a crucial role in maintaining motivation and continuing the learning process. The Instructional Material Motivational Survey (IMMS), developed by [37,44], is an instrument designed to diagnose motivation based on this model.

While originally formulated for face-to-face teaching, the ARCS model and its accompanying instrument have found utility in various instructional contexts involving technology-mediated training, such as virtual teaching with video, podcasts, MOOCs, computer-assisted teaching, distance learning, interactive video, computer animations, and simulations [45–51]. Recently, it has been increasingly employed in training actions involving immersive reality (IR) [3,52–55].

Building upon the aforementioned concepts, the primary objective of this research is to provide reliable insights into the training of healthcare professionals in the field of Health

Sciences, with a specific focus on the use of immersive reality (IR) and 360° video in the initial training of doctors at the University of Seville. The study aims to examine the impact of these technologies on various aspects of medical education, including student motivation, learning outcomes, and the development of medical skills. By assessing motivation levels among students and analyzing performance tests, the research seeks to determine the effectiveness of IR and 360° video in enhancing experiential learning and creating more engaging and realistic learning environments. Furthermore, the study aims to contribute valuable insights to the existing body of knowledge in medical education, specifically within the context of Seville, while highlighting the potential benefits and implications of incorporating these technologies into the training of healthcare professionals.

## 3. Methods

### 3.1. Objectives

The research was carried out during the 2021–2022 academic year with different objectives in mind. This article highlights the following ones among them:

1.  Examining the degree of motivation perceived through the IMMS that the use of learning objects in IR, designed in a specific way through the use of "hot spots" to offer additional information (computer points, incorporation of texts, incorporation of video clips . . . ). In the experience, immersive reality has been understood as an immersive reality that combines augmented reality (markers) and 360° video (recording of situations in the hospital).
2.  Analyzing the degree of attention, relevance, trust, and satisfaction analyzed through the IMMS that the use of learning objects in IR, designed in a specific way through the use of "hot spots" to offer additional information (computer points, incorporation of texts, incorporation of video clips...).
3.  Analyzing whether exposure to objects produced in immersive reality facilitates the acquisition of knowledge.

### 3.2. Description of the Experience

The study focused on the field of surgery within the Medicine Degree program, where students are required to undergo a practical period in the surgical area. This practice period aims to familiarize students with surgical techniques and skills while exposing them to the demanding and potentially hostile environment of the operating room.

Considering that third-year students are experiencing their first professional encounter in a hospital setting, it is common for them to feel stressed and overwhelmed. To alleviate some of this stress, it would be beneficial to provide students with a preview of the scenarios they will encounter during their practical period. This preview can help them develop self-confidence and a sense of orientation.

The three scenarios that students will face include surgery consultations, the "clean" area where handwashing is conducted before surgery, and the operating room itself. Three immersive-reality learning objects were developed to facilitate learning and understanding of these scenarios. These learning objects incorporated 360° videos to provide a comprehensive explanation of each scenario.

The three learning objects are as follows:

1.  Consultation scenario: This object demonstrates how students should interact with patients during consultations, particularly when communicating diagnoses (Figure 2).
2.  Handwashing scenario: This object guides students through the proper handwashing procedure conducted before surgical interventions (Figure 3).
3.  Operating room scenario: This object illustrates how students should navigate and conduct themselves within an operating room (Figure 4).

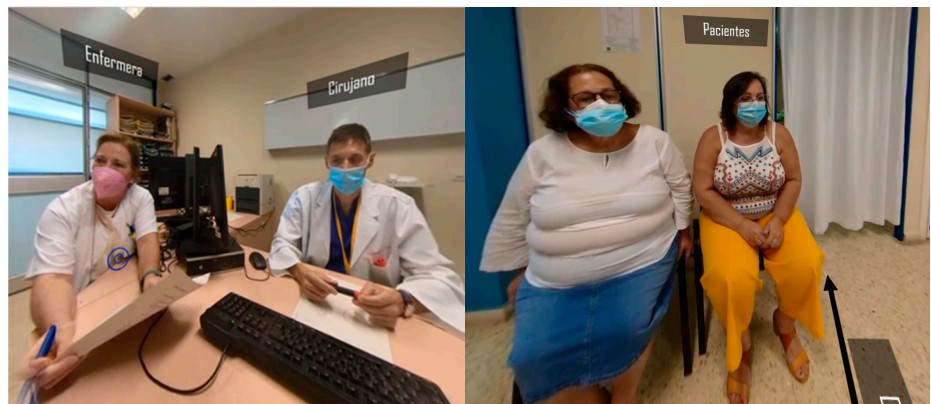

**Figure 2.** Images object of learning "query". Note: "Enfermera" = Nurse; "Cirujano" = Surgeon; "Pacientes" = Patients.

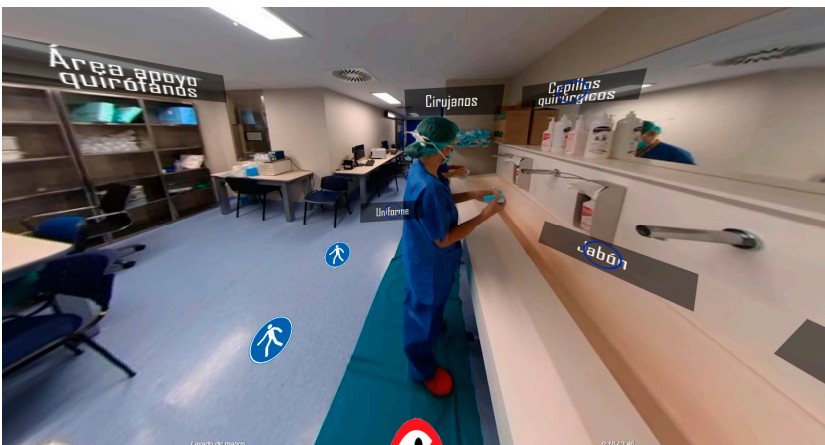

**Figure 3.** Images object of learning "hand washing". Note: "Área apoyo quirófanos" = Operating room support area; "Cirujanos" = Surgeons; "Cepillos quirúrgicos" = Surgical brushes; "Jabón" = Soap.

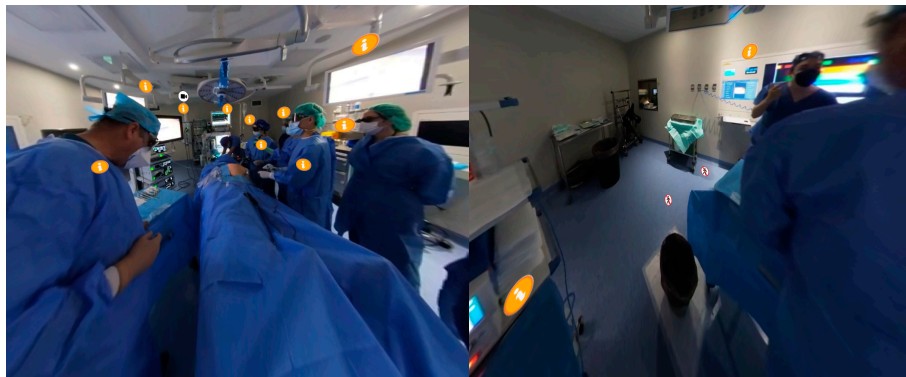

**Figure 4.** Images learning object "operating room".

It is important to note that these three learning objects are classified as non-immersive, according to [56]. The following links provide access to the observation of the three learning objects produced: https://ra.sav.us.es/rv/imagen/intro/ (accessed on 12 July 2023).

It is important to highlight that the learning objects incorporated "hot spots," which were interactive elements containing additional information relevant to the learning objectives or specific details depicted in video clips or images.

Furthermore, at the conclusion of the interaction within the learning objects, an informative poster appeared, prompting the students to complete a questionnaire. This

questionnaire aimed to assess their comprehension and retention of the information presented throughout the different learning objects.

The number of "hot spots," the resources utilized, and the type of information provided varied across the different learning objects. For a comprehensive list of these elements, please refer to Table 1.

**Table 1.** "Hot spots" and resources incorporated in the three objects produced.

| Resources | Operating Room | Handwashing | Consultation |
|---|---|---|---|
| Informational Hot Spots | 14 (type of presentation and description) | 5 (information expansion typology) | 2 (one of evolution and with five informative posters calling for attention and one informative). |
| Video clips | 1 | | |
| Location zone points | 2 | 5 | 2 |

To implement the different learning objects, the following resources and programs were utilized: a 360° one R camera, the editing software provided by the camera manufacturer, Adobe Premiere software for video editing, and the KRPANO program for creating the interactive "hot spots". Once the objects were created, they were uploaded to a server for accessibility.

Students were given the respective URLs of the learning objects and were able to interact with them within the classroom setting. After engaging with the objects, students completed the questionnaire provided at the end of each training session to evaluate their learning experience.

### 3.3. Instrument

Two different instruments were employed to gather data: the IMMS [37] for assessing motivation and a custom test for measuring information acquisition.

The IMMS [44] questionnaire consisted of 35 Likert-scale items, ranging from 1 (Extremely unlikely/disagree) to 7 (Extremely likely/agree). It assessed the four dimensions of the ACRS model [37]: attention, confidence, relevance, and satisfaction. It is important to note that all values were treated as interval data. The instrument's reliability has been established in previous studies conducted by our research group in the field of education, with Cronbach's alpha ranging from 0.898 to 0.960 [3,52–54,57,58]. Given the consistent reliability of the instrument, there was no need to reevaluate it for the present study.

The IMMS questionnaire was administered online after students interacted with the immersive reality learning objects.

For performance analysis, a custom instrument comprising ten items of various types, such as multiple choice, fill in the blanks, and true or false, was developed.

### 3.4. Sample

The study included a sample of 136 students enrolled in the Fundamentals of Surgery and Anesthesiology course during their third year of the Medicine Degree at the University of Seville.

It is important to note that the sample for this study was selected using a non-probabilistic approach, specifically a convenience or causal sampling method [59]. The selection of participants was based on the accessibility and availability of students within the researcher's reach.

With a total population of 560 medical students enrolled at the University of Seville, the sample size of 136 students yields a margin of error of 7% with a 95% confidence interval.

### 4. Results

It is worth emphasizing that the overall mean score obtained in the instrument was 5.41, with a standard deviation of 0.695. This indicates that students perceived a high level

of motivation when participating in the immersive reality experience. The low standard deviation suggests a certain degree of consistency in the scores reported by the students.

Table 2 presents the mean scores and standard deviations for each item comprising the instrument. To correctly interpret the results, it is important to note that the scale ranged from a minimum value of 1 to a maximum value of 7.

**Table 2.** Mean scores achieved in the different items (note: C = Confidence, A = Attention, S = Satisfaction, and R = Relevance).

| Items | M | SD |
|---|---|---|
| 1. At first glance, I had the impression that the activities with the object in IR would be easy for me. (C) | 5.32 | 1.281 |
| 2. There was something interesting in the IR materials that caught my attention. (A) | 5.98 | 1.151 |
| 3. The contents offered in IR are more difficult to understand than I would like (C) | 3.02 | 1.523 |
| 4. After reading the statements, it was clear to me what I was going to learn working with the simulator. (C) | 2.96 | 1.333 |
| 5. When completing the activities of the object in IR, I had the satisfactory feeling of having achieved the objectives. (S) | 5.49 | 1.211 |
| 6. It has become clear to me how the content of the object's activities in IR is related to things I already knew (R) | 5.69 | 1.152 |
| 7. The information was so much that it was difficult for me to grasp and remember the important points (C) | 4.91 | 1.580 |
| 8. The technology of the object in IR caught my attention and was attractive (A) | 5.99 | 1.256 |
| 9. The activities, images and videos with the IR object have shown me how this teaching material could be important for some people (R) | 5.97 | 1.180 |
| 10. Successfully completing this material was important to me (R) | 5.91 | 1.244 |
| 11. The visual quality of the object in IR helped me keep my attention (A) | 5.79 | 1.285 |
| 12. The IR object material was so abstract that it was hard to keep my attention on it (A) | 5.56 | 1.470 |
| 13. As I worked on the object in IR, I felt sure that I would be able to learn the contents (C) | 5.41 | 1.208 |
| 14. I have enjoyed working with the object in IR so much that I would like to know more about it (S) | 5.31 | 1.406 |
| 15. The images, videos and texts of the object in IR are unattractive (A) | 5.72 | 1.315 |
| 16. The contents worked on in the IR object are relevant to my interests (R) | 5.93 | 1.034 |
| 17. The way the information is organized in the object in IR helped me keep my attention (A) | 5.65 | 1.255 |
| 18. Explanations or examples of how the knowledge learned in the object can be used in IR (R) were included | 5.32 | 1.428 |
| 19. The activities with the object in IR were too difficult (C) | 5.54 | 1.366 |
| 20. The information presented in the object in IR has things that stimulated my attention (A) | 5.81 | 1.177 |
| 21. I really liked studying this content (S) | 5.72 | 1.286 |
| 22. The number of repetitions in the activities bored me sometimes (A) | 4.81 | 1.448 |
| 23. I have learned some things with the object in IR that I found surprising or unexpected (A) | 5.10 | 1.368 |
| 24. After working for a while with the object in IR, I felt confident that I was going to be able to pass a test on the content presented (C) | 5.00 | 1.372 |

**Table 2.** *Cont.*

| Items | M | SD |
|---|---|---|
| 25. The contents and tasks of the object in IR were not relevant to my needs, because I already knew more about the content (R) | 5.11 | 1.438 |
| 26. The achievements made on the object in IR helped me feel rewarded for my effort (S) | 4.96 | 1.561 |
| 27. The variety of audiovisual material helped keep my attention (A) | 5.67 | 1.300 |
| 28. The material presented in the IR object was boring (A) | 5.55 | 1.514 |
| 29. I could relate the contents worked on in the IR object with the things I have seen, done or thought about before (R) | 5.86 | 1.301 |
| 30. There is so much content in the object in IR that it is irritating (A) | 5.85 | 1.305 |
| 31. It felt good to successfully complete the item in IR (S) | 5.85 | 1.179 |
| 32. The contents worked on the object in IR will be useful to me (R) | 6.02 | 1.071 |
| 33. I haven't really been able to understand the object material in IR (C) | 5.76 | 1.518 |
| 34. The good organization of the material in the IR object helped me feel confident that I was going to learn its content (C) | 5.49 | 1.253 |

Only two items on the questionnaire scored below the mean value of the scale, which was 4. These items were "After reading the statements, it was clear to me what I was going to learn working with the simulator" (2.96) and "The contents offered in IR are more difficult to understand than I would like" (3.02). It is important to note that out of the 35 questionnaire items, only these two were rated below 3.5, which accounts for 5.71% of the student responses. Both of these items belonged to the "trust" dimension.

On the other hand, several items obtained the highest scores, including "The images, videos, and texts of the object in IR are unattractive" (5.72), "I really liked the study of this content" (5.72), "I have not been able to understand the material of the object in IR" (5.76), "The visual quality of the object in IR helped me to keep my attention" (5.79), "The information presented in the object in IR has things that stimulated my attention" (5.81), "There is so much content in the IR object that it is irritating" (5.85), "It felt good to successfully complete the IR object" (5.85), "I could relate the contents worked on in the IR object with the things I have seen, done, or thought about before" (5.86), and "Completing this material successfully was important to me" (5.79). These high-scoring items were primarily from the "attention" dimension, followed by "satisfaction" and "relevance".

Table 3 provides the mean scores and standard deviations achieved in the different dimensions established in the Keller IMMS test.

**Table 3.** Mean scores and standard deviations achieved in the IMMS dimensions.

| Dimensions | M | SD |
|---|---|---|
| Confidence | 4.82 | 0.518 |
| Attention | 5.62 | 0.855 |
| Satisfaction | 5.47 | 1.011 |

Once again, it was observed that the average scores in all dimensions exceeded the mean value of the scale, indicating that students perceived high levels of motivation across the board. The standard deviations were relatively low, suggesting a certain level of agreement in the students' responses. Notably, the dimension related to the "relevance" of the activity received the highest rating among the students.

As previously mentioned in the research objectives, one of the aims was to assess the students' interaction with the three produced objects.

Turning to performance, Table 4 provides the average scores and standard deviations for each performance test conducted, along with the overall average and standard deviation across all tests.

**Table 4.** Average scores achieved in performance.

|  | Operating Room | Handwashing | Consultation | Total |
|---|---|---|---|---|
| M | 9.5772 | 7.8088 | 8.0294 | 8.4718 |
| SD | 0.98860 | 1.61252 | 1.28778 | 0.74468 |

As can be observed, the average scores achieved by the students denoted that the objects produced served to acquire the different contents that were presented to them.

## 5. Conclusions

This research endeavors to provide reliable findings concerning the training of professionals in the field of Health Sciences, specifically focusing on the use of immersive reality (IR) and 360° video in the initial training of doctors at the University of Seville.

The study examined the perceived motivation levels of students using the IMMS instrument, and the results corroborate those found in related research. Evidence supports the positive effect of IR on learning in general [9,12,13], including improvements in students' sense of physical presence, learning experience, and satisfaction when participating in training experiences under this modality [15–18,60]. Notably, motivation levels have been shown to increase with the use of IR in various contexts, including the field of Health Sciences [18,22,24,25,61,62].

Furthermore, the study investigated whether exposure to immersive reality objects facilitates knowledge acquisition. Analysis of the performance tests and existing research supports the significance of IR and 360° video in enhancing experiential learning [16,20–22]. Consequently, it can be concluded that the use of these technologies promotes the development of medical skills [62] and creates more active and participatory training scenarios for students [11,63].

The study also revealed the significance of using "hot spots" within the design of learning objects, aligning with previous research [27,28,64,65]. Incorporating these interactive elements enhances the learning experience and contributes to better understanding and engagement.

Implementing immersive reality (IR) and 360° video in medical education offers significant advantages. These technologies can create more engaging and realistic learning environments, simulating real-world medical scenarios. The hands-on and experiential approach prepares students for the challenges they will face in their professional careers. Moreover, the study highlights the positive impact of IR on student motivation and engagement, increasing their interest in learning and their active participation in the educational process. This heightened engagement can lead to improved knowledge retention and a deeper understanding of medical concepts.

Additionally, the research emphasizes the facilitation of medical skill development through the use of IR. By providing opportunities for students to practice and apply their knowledge in realistic scenarios, these technologies enable the refinement of clinical skills, decision-making abilities, and critical thinking. This, in turn, leads to the cultivation of competent and confident healthcare professionals [28].

The study aligns with similar research in the field of medical education, where the use of IR technology has gained traction. Its application extends to areas such as rehabilitation, disability management, surgical training, therapy for psychological diseases, and pain management [66]. By liberating learning from traditional classrooms, IR is revolutionizing medical education. It allows learners to apply their knowledge in practical scenarios and learn from their mistakes. IR emphasizes competency enhancement and promotes autonomous, blended learning, aligning with the expectations of today's learners [67].

A key novelty of this study lies in its specific context—focused on the University of Seville. By conducting the research within this specific setting, valuable insights into the use of IR and 360° video in the initial training of doctors at this university are provided. This context-specific approach offers localized perspectives and allows for a deeper understanding of the benefits and implications of incorporating these technologies in Health Sciences education.

Another noteworthy aspect is the incorporation of immersive reality with "hot spots," which has captured the attention of medical students worldwide. These hot spots serve as focal points within the immersive experience, enabling students to direct their attention to specific aspects of the virtual environment, thereby enhancing engagement and understanding of complex medical concepts. The use of immersive reality with hot spots offers a transformative and dynamic formative experience that surpasses traditional learning methods, providing students with a truly immersive and interactive learning environment [52].

The study contributes to the existing body of knowledge by examining student motivation using the IMMS instrument. The results not only support previous findings but also emphasize the positive effect of IR on learning, physical presence [13,16], and student satisfaction in formative experiences [28,42]. Additionally, the study highlights the motivation levels of students in the field of Health Sciences, further reinforcing the significance of incorporating IR and 360° video in medical education. These findings align with [31], who assert that immersive learning generates high levels of motivation among university students. Similar studies conducted outside of university settings, such as [46,53], also support these assertions. In this context, the study by [66] links motivation to "engagement" and the "applied practicality" of immersive experiences.

Furthermore, the research investigates the effectiveness of immersive reality in facilitating knowledge acquisition and the development of medical skills. By analyzing performance tests and building upon existing research, the study provides evidence of the significance of IR and 360° video in enhancing experiential learning and creating more active and participatory training scenarios. The emphasis on the design of learning objects with "hot spots" aligns with previous findings and underscores the importance of this approach in promoting learning and motivation.

However, the researchers acknowledge the limitations of conducting the study in a specific context and, therefore, suggest the need for further research in different universities and non-university training spaces to replicate the experience and explore other critical variables that may influence the integration of emerging technologies. This call for future research opens new avenues for investigation and expansion of knowledge in the field.

One significant limitation of the study is the absence of a comparison with traditional learning methods that do not incorporate technology. Without including a control group or contrasting the results with non-technology-enhanced experiences, it becomes challenging to attribute the observed improvements in motivation and performance solely to immersive reality. A recommended future direction is to employ a contrastive quasi-experimental design to overcome this limitation. In this design, one group of students would undergo immersive reality training, while another group would receive comparable training without the use of technology. This comparative approach would provide a more comprehensive understanding of the unique benefits offered by immersive reality in medical education and facilitate evidence-based decision-making for its implementation.

In conclusion, this research has demonstrated that the use of immersive reality (IR) and 360° videos is a technology that fosters motivation and can be effectively incorporated into university education, evoking satisfaction and success. These resources can be utilized in teaching to capture attention, interest, and curiosity in learning. The study confirms the positive impact of IR on motivation, learning outcomes, and the development of medical skills. By creating engaging and realistic learning environments, IR enables students to acquire knowledge and practice essential skills in practical scenarios.

Furthermore, the study contributes to the existing body of knowledge by providing valuable insights specific to the University of Seville and the incorporation of immer-

sive reality with "hot spots." The localized perspective offers a deeper understanding of the benefits and implications of integrating these technologies in the field of Health Sciences education.

It is evident that immersive reality, coupled with interactive elements such as hot spots, has the potential to transform medical education by bridging the gap between theory and practice. It provides students with immersive, interactive, and experiential learning opportunities, preparing them to become competent healthcare professionals. However, further research is needed to replicate and expand upon these findings, explore additional variables, and incorporate comparative studies to fully understand the unique advantages of immersive reality in medical education. With continued investigation, immersive reality can be effectively harnessed to enhance medical training, improve student outcomes, and meet the evolving needs of healthcare education.

Moreover, the implications of this research extend beyond the field of medical education. The findings highlight the potential of immersive reality and 360° video in enhancing learning experiences and promoting engagement in various disciplines. The ability to create realistic and immersive environments can benefit fields such as engineering, architecture, and psychology. The integration of immersive technologies can revolutionize the way education is delivered, making it more interactive, practical, and engaging for students across different domains.

Additionally, the study emphasizes the importance of instructional design and the incorporation of interactive elements within immersive reality experiences. The use of "hot spots" and other interactive features enhances the learning process by directing students' attention to specific aspects of the virtual environment. This targeted focus allows for deeper exploration and understanding of complex concepts, thereby facilitating knowledge acquisition and skill development.

Furthermore, the research highlights the need for ongoing professional development and training for educators to effectively integrate immersive reality technologies into the curriculum. Educators must become familiar with the capabilities of these technologies, understand their potential impact on learning outcomes, and learn to design and implement immersive experiences that align with educational objectives. Investing in educator training and support will ensure the successful integration of immersive reality in educational settings.

In conclusion, this research provides valuable insights into the integration of immersive reality and 360° video in medical education, showcasing their potential to enhance motivation, learning outcomes, and the development of medical skills. The study contributes to the existing body of knowledge by highlighting the benefits of immersive technologies and emphasizing the importance of instructional design and educator training. As the field of immersive reality continues to evolve, further research is needed to explore its applications in diverse educational contexts and to investigate its long-term impact on student engagement, knowledge retention, and professional competence. With continued research and innovation, immersive reality has the potential to reshape education and create transformative learning experiences for students in various disciplines.

**Author Contributions:** Conceptualization, J.C.-A.; methodology, A.P.-R.; software, A.P.-R.; validation, F.D.-L.-P.-D.-J. and J.B.-O.; formal analysis, J.C.-A.; investigation, F.D.-L.-P.-D.-J.; resources, F.D.-L.-P.-D.-J.; data curation, A.P.-R.; writing—original draft preparation, J.C.-A.; writing—review and editing, J.B.-O. and A.P.-R.; visualization, F.D.-L.-P.-D.-J.; supervision, F.D.-L.-P.-D.-J. and J.B.-O. All authors have read and agreed to the published version of the manuscript.

**Funding:** This research received no external funding.

**Institutional Review Board Statement:** Not applicable.

**Informed Consent Statement:** Informed consent was obtained from all subjects involved in the study.

**Data Availability Statement:** https://grupotecnologiaeducativa.es/ (accessed on 1 July 2023).

**Conflicts of Interest:** The authors declare no conflict of interest.

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
