# Peer review of "Technology-Enhanced Learning in Health Sciences: Improving the Motivation and Performance of Medical Students with Immersive Reality"

_applsci, doi:10.3390/app13148420_

Round 1

Reviewer 1 Report (Previous Reviewer 2)

The authors aims to study the motivation and performance levels obtained by students of the Degree in Medicine of the University of Seville after experiencing a session with IR. The work is interesting but the authors should address some important points.

According to the authors, "Numerous studies suggest that Immersive Reality (IR) is an educational technology with a great potential in the field of Health Sciences. Its integration allows to increase motivation and academic performance of students.". The authors base their work on previous ones to propose an experiment and conclude the same results: "In short, this research has shown that IR and 360° videos is a technology that awakens true motivation in students, and therefore can be perfectly used in university education to awaken satisfaction towards a successful achievement, not arousing any suspicion for its incorporation. Therefore, they can be resources to use in teaching to arouse attention, interest and curiosity towards learning.".

The critical point in my analysis here is the lack of novelty of this work. This is needed for acceptance of the paper. If the authors based the experiment in previous knowledge and found results that differed from previous ones, it would be an opportunity to discuss them. But the experiment was performed without changing much of what has been done in previous works, and the user analysis is based on conventional techniques, so that no new results are obtained.

In order to have the paper accepted, the authors must wok in this direction: they need to explicitly compare the findings of their current paper to previous ones and clearly show how it improves state of the art. I've noticed the effort of the authors in including new references to the paper and other material, but the "novelty effect" is still missing.

More general comments and minor errors are listed as follows.

The goal of the work is not described in the introduction section.

The objectives of the work are only defined inside the "methods" section. The "methods" section should explain "how", not "what".

"this article highlight" -> "this article highlights"

Author Response

Reviewer 2 Report (Previous Reviewer 1)

Mean scores and standard deviations cannot be regarded as reliable statistical tests (it was my plea in the first revision). Therefore, the conclusion, the research has shown that technology awakened students' motivation is a kind of abuse as it is unsupported by statistical evidence.

Besides, the study group (136 students) is rather small to draw sound conclusions.

Author Response

Reviewer 3 Report (New Reviewer)

This paper describes the result of questionnaire surveys and test scores after the class experience of the immersive technology to show the effects on students’ motivation.

The technology will be promised and well literature in this paper, although this paper has many weak points to support their results.

(1)  There is no comparison between the experimental setting and the control setting. Therefore, the author cannot describe the superiority of the technologies, generally.

(2)  The statistical analysis is insufficient: why is the mean-value 3.5? The average value between 1 and 7 should be 4. The value of the Likert-scale questionnaire is an ordinal scale, so the author should reconsider the statistical analysis. There is no statistical test to describe the reliability test of the answer values. The interpretations of the results are too naive to support their objects.

(3)  The ARCS model is the learning process model in education research. Why can we use the model to check the degree of the learner’s motivation?

Above the comments (or reasons), the reviewer is sorry to judge the rejection of this paper.

Round 2

Reviewer 1 Report (Previous Reviewer 2)

Dear authors, thank you for the effort considering the past comments. I'm satisfied regarding the point about the methods section, but other (critical) points remain open.

I'm still concerned regarding thr novelty of the work. Please make this very clear in the text as applying well known techniques to a different location does not necessarily brings enough novelty that justifies the publication.

Please be explicit on the contributions of the proposed work compared to previous ones.

Please perform a careful review on the newly added parts. For instance, the conclusion section has some repeated information in it (regarding the limitations of the work).

Author Response

Reviewer 3 Report (New Reviewer)

Minor comments:

1. Statistics

1) I cannot understand the sentence, yet.

"There are only two items that did not exceed the mean value of the scale, which was 3.5, specifically."

(1,2,3,4,5,6,7) <= If the minimum value is 1 and the maximum value is 7, then the expected mean value of this scale may be 4.0. Off course, if the minimum value becomes 0, then the mean value of 3.5 was acceptable. (?0,1,2,3,4,5,6,7 <= eight values)

2) I have pointed out that the value of the Likert-scale questionnaire is an ordinal scale, ordinary. However, your paper treats the value as an interval scale. It is better to mention that you treat the value as the interval scale.

If you could not understand my comments, then please check "Level of measurement." in statistics.

2. Motivation

You answer "In this case, the objective of the article is not to place technology above."

To my knowledge, the researches on educational technology take care of the increase in students' motivation caused by using new technology, so I commented. I hope your understanding and consideration to write some limitations of your research.

Author Response

This manuscript is a resubmission of an earlier submission. The following is a list of the peer review reports and author responses from that submission.

Round 1

Reviewer 1 Report

The most serious objections to the content of the paper relate to the break of the statistical tests performed to compare the results of achieving the learning objectives of students from the group of those using VR technology compared to students who did not use the technology.

The text of the paper is edited carelessly. It needs serious improvement because it contains numerous grammatical errors and typos. 

For example, there is "...[1] point out" instead of "...[1] points out" or "...As [3] point out" instead of "As [3] points out", etc. 

There are also missing comma(s), and typos, e.g., "...and it provides a sensation -sation of total" or " ...head-mounted display (HMD, for its acronym in Spanish). in English)..."

The examples of typos: "Inmersive" instead of "Immersive" (in the title), "Whearas" instead of "Whereas", "Objetives" instead of "Objectives", "Numerous studies have reported ... in general [9, 12-14]; o specifically " (unnecessary letter "o" before "specifically"), etc.

I am also interested in why you use "he" instead of "he/she" or "they" when talking about a person's experiences sth. 

Therefore, it is highly recommended that the paper be reviewed by a language expert or - at least - checked using the Grammarly service.

Reviewer 2 Report

The authors study the level of motivation and performance obtained by the students of the Degree in Medicine of the University of Seville after experiencing a session with Virtual Reality and 360° Video. The idea is interesting, but contrary to what authors claim, this is not new. The authors must also be careful with the paper writing and perform a general review, as lots of minor errors were found and are listed in sequence.

General comment regarding citations in the text: whenever the citation is part of the sentence (e.g. "[37] indicates that it refers "), please use the author name followed by the reference number (e.g. "Keller [37] indicates that it refers "). When the reference is not critical to the sentence (it can be removed without affecting the sentence understanding), just using the reference number is ok (e.g. " its scientific production has increased [7]. ").

The authors say that "Numerous studies have reported a positive effect of VR on learning" and "We also find a large volume of research that points to the significance of its use to enhance experiential learning". That being said, what contributions are brought by the work compared to what already exists in the current state of the art? This must be very clear so the paper can bring novelty and be accepted.

"In this sense, the research allows us to discover a reality that has not been explored until now: the use of VR and 360º video in the initial training of doctors." -> this is not completely true. The authors do not cite some important work that already make use of 360 videos for training doctors. Some of them are listed as follows and should make part of the text and also be compared to the proposed work:

- "The Potential of 360° Virtual Reality Videos and Real VR for Education—A Literature Review"

- 360° videos in education – A systematic literature review on application areas and future potentials

- 360° virtual reality video for the acquisition of knot tying skills: A randomised controlled trial

- Using 360-degree video for teaching emergency medicine during and beyond the COVID-19 pandemic

- Educational 360-Degree Videos in Virtual Reality: a Scoping Review of the Emerging Research

- 360° vision applications for medical training

- Implementing 360-Degree Simulation Training During Psychiatry Placement Inductions: A Mixed Methods Training Evaluation

In summary, the authors must clearly present the novelty that their work brings and directly compare it to previous works so the paper can be accepted. Unfortunately, this is not present in the current version of the paper.

More general comments and minor errors are listed as follows.

It is not common to use "." in the paper title. Suggestion for new paper title: "Technology-Enhanced Learning in health sciences: motivation and performance of medical students with the use of Inmersive Reality and 360º video"

"this research is presented that tries to study" -> "this research tries to study"

"360º" -> please correct the "degree" symbol. it should not have the line below "o"

"Mixed Reality (MR) being a combination of both technological formats. " -> please rewrite

" sensation -sation of total immersion;" -> " sensation of total immersion;"

" in English) [4]." -> ?

" as presented in the scheme presented" -> please rewrite

"Whearas," -> "Whereas,"

"o specifically" -> "or specifically"

"360º video," -> "360° video,"

"360º video" -> "360° video"

"360º video," -> "360° video,"

"360º video" -> "360° video"

"moment have" -> "moment, have"

"incorporating 360º videos" -> "incorporated 360° videos"

"creating of the" -> "creating the"

"Completing the questionnaires at the end of each interaction, as indicated above." -> this sentence seems incomplete, please rewrite it

"3.3. Instrument." -> "3.3. Instrument"

"[44], consisted" -> "[44] consisted"

"Indicate that the instrument has already been used" -> please rewrite

" analyzed. of its reliability index " -> ?

" true or false..." -> " true or false, etc."

"(TO)" -> ?

In Table 2, please replace "," by "." as decimal separator

In Table 3, please replace "," by "." as decimal separator

In Table 4, please replace "," by "." as decimal separator

"As we already stated in the objectives, one of them referred to knowing if the students, in the interaction with the three objects produced, acquired" -> ?

"360º video" -> "360° video"

"experience. -teaching and" -> ?

"360º video" -> "360° video"

"360º video" -> "360° video"

"in different university" -> "in different universities"

"360º videos" -> "360° videos"

" achievement. Not" -> " achievement, not"

Round 2

Reviewer 2 Report

Thank you for submitting a revised version of the manuscript. However, I was not satisfied with some of the answers given by the authors and listed in sequence.

"The study has been thoroughly reviewed by a native peer reviewer specializing in scientific texts. " -> It was still possible to find writing errors in the text, even in the title of the paper. This makes me wonder how accurate was this text review...

Some examples of the current errors are:

"Technology-Enhanced Learning in Health Sciences: Motivation and performance of medical students with the use of Immersive Reality." -> please remove the final "."

"conceptualization, [1]" -> "conceptualization [1], "

"their part, [2]" -> "their part [2],"

" as [5] it has been pointed out," -> " as [5] has been pointed out,"

"on VR ," -> "on VR,"

"students, are" -> "students are"

"the two different instruments" -> "Two different instruments"

"(TO)" -> the authors have still not defined the meaning of "TO"

"produced, acquired" -> ?

"Response: These studies have been included. We thank the reviewer for this contribution, since it significantly improves the scientific quality of the study." -> How were the seven studies were included in the text if the original one had 64 references and the new version has 66? What did I miss here?

"In this sense, the research allows us to discover a reality that has not been explored until now: the use of VR and 360º video in the initial training of doctors." -> What I meant by "this is not completely true is that that are already other works dealing with the use of VR and 360 degree videos for training doctors. Making this sentence specific (regarding the training of doctors in Seville) simply does not solve the problem. The authors must be explicity on the novelty brought by this paper and this is not the case.

The authors propose a case study including the use of VR and 360 degree videos to train doctors, but where is the novelty in it? Besides that, they do not compare their work with previous ones, making it difficult to compare and notice the value of the proposed work against others. I strongly recommend the authors to re-evaluate their work, improve it and then submit it again in a new opportunity on the future.

"The practical implications of the study have been expanded trying to contemplate what has been said in this commentary." -> this is yet not clear in the text. Unfortunately for this reason, I am now recommeding a reject.

Comments regarding quality were already mentioned before.

Round 3

Reviewer 2 Report

Thank you for the quick responses. My point still regards the novelty of the work. In the conclusion section, the authors still mention "In this sense, the research allows us to discover a reality 279 that has not been explored until now: the use of IR and 360° video in the initial training of 280 doctors at the University of Seville."

The authors should justify their work based on the proposed novelty, or based on the new findings that the work brings, instead of the place the research was realized. According to the authors, one may reproduce the exact same experiment on a different university/hospital and publish a new paper. Research papers must improve state of the art  and bring novelty, so please make this topic clear, as requested before.

Some works that can be used for comparing and highlighting the novelty of this one are:

- https://www.ncbi.nlm.nih.gov/pmc/articles/PMC5622235/

- https://www.ncbi.nlm.nih.gov/pmc/articles/PMC6798020/
